# WHAT CAN MULTI-CLOUD CONFIGURATION LEARN FROM AUTOML?

## ABSTRACT

Multi-cloud computing has become increasingly popular with enterprises looking to avoid vendor lock-in. While most cloud providers offer similar functionality, they may differ significantly in terms of performance and/or cost. A customer looking to benefit from such differences will naturally want to solve the multi-cloud configuration problem: given a workload, which cloud provider should be chosen and how should its nodes be configured in order to minimize runtime or cost? In this work, we consider this multi-cloud optimization problem and publish a new offline benchmark dataset, MOCCA, comprising 60 different multi-cloud configuration tasks across 3 public cloud providers, to enable further research in this area. Furthermore, we identify an analogy between multi-cloud configuration and the selection-configuration problems that are commonly studied in the automated machine learning (AutoML) field. Inspired by this connection, we propose an algorithm for solving multi-cloud configuration, CloudBandit (CB). It treats the outer problem of cloud provider selection as a best-arm identification problem, in which each arm pull corresponds to running an arbitrary black-box optimizer on the inner problem of node configuration. Extensive experiments on MOCCA indicate that CB achieves (a) significantly lower regret relative to its component black-box optimizers and (b) competitive or lower regret relative to state-of-the-art AutoML and multi-cloud methods, whilst also being cheaper and faster.

## 1 INTRODUCTION

We are currently living in the era of cloud computing, in which cloud providers compete to offer computing resources, including servers, storage and managed software, to businesses and consumers via the internet. Cloud is a high growth segment: according to Gartner (2021), worldwide end-user spending on public cloud services is forecast to grow 23.1% in 2021 to total $332.3 billion. From an enterprise perspective, there are many benefits to utilizing cloud services including, but not limited to: increased flexibility, reduced costs, improved security and rapid scaling.

Coupled to the rise of cloud is the growing popularity of containers and container orchestration platforms. A container image (e.g., Docker) is an executable package of software that includes everything needed to run an application: code, runtime, system tools, system libraries and settings (Soltesz et al., 2007; Merkel, 2014). At runtime, container images become containers and can be deployed quickly and reliably from one computing environment to another, making them well-suited to the world of cloud computing. Many of today's workloads, such as machine learning (ML) training, are distributed in nature and require the deployment and management of multiple containers, all working together in parallel. Container orchestration platforms such as Kubernetes (Brewer, 2014) enable running such systems resiliently, providing features like auto-scaling and failure management.

Most cloud providers now offer a Kubernetes service, meaning that they require only a few commands to create a Kubernetes cluster, and the user can specify the desired number of nodes and how each node should be configured (how many CPUs, memory, etc.). Given a Kubernetes cluster, one can easily run a complex distributed workload, whilst remaining agnostic to which cloud provider is providing the resources in the backend. This level of abstraction presents the savvy business or consumer with a new optimization opportunity, which we will refer to as the *multi-cloud configuration problem*. Namely, given a workload, which cloud provider should be selected, and how should the nodes in the cluster be configured, in order to minimize runtime or cost?

In practice, this optimization problem is of particular interest to businesses with complex, distributed workloads that must be run repeatedly. For instance, consider a business with a ML-based recommendation engine that must be trained, in a distributed fashion, once every hour. By solving the multi-cloud configuration problem, the business will be able to exploit price/performance differences that may exist between cloud providers to improve their bottom-line. However, any optimization algorithm will involve making a certain number of evaluations (e.g., running the workload on a particular cloud provider, with a particular node configuration), and these evaluations will themselves incur a certain dollar cost. For this reason, effective algorithms with fast convergence are of particular interest in this domain, and are the primary focus of this paper.

The main contributions of our work are as follows:

- We formally introduce the multi-cloud configuration problem and draw an interesting connection with the selection-configuration problems that are commonly studied in the automated machine learning (AutoML) field.
- Inspired by AutoML approaches, we propose a best-arm identification algorithm, Cloud-Bandit, for solving the multi-cloud configuration problem.
- We present a new dataset, MOCCA, for offline benchmarking of optimization algorithms applied to 60 multi-cloud configuration tasks, across 3 public cloud providers.
- Extensive experiments on MOCCA show that CloudBandit outperforms many generic black-box optimizers and state-of-the-art selection-configuration algorithms.

## 2 PROBLEM STATEMENT AND BACKGROUND

### 2.1 MULTI-CLOUD CONFIGURATION AS OPTIMIZATION

The runtime/cost of a workload in the cloud depends on many factors, such as the number of nodes used, the vCPU count, the amount of memory allocated to each node, the manufacturer/generation of the backend CPU, the network on which the nodes communicate and the region in which the nodes are deployed. While some of these parameters (e.g., vCPU count and amount of memory) are, on the surface, common across different cloud providers, most providers do not give the option to set these parameters freely. Instead, one must select from a set of available VM categories (often parameterized into sub-categories like *family*, *type* or *size*) that define which CPU will be used, how many vCPUs and how much memory will be assigned to the node, as well as what network interfaces are used. These (sub-)categories are significantly different across cloud providers, making it difficult to construct a multi-cloud optimization problem over parameters that are common across all cloud providers (e.g., vCPU count) without introducing complex constraints.

Instead, it is much easier to view the multi-cloud configuration problem as an optimization over a hierarchical domain, where the domain for each cloud provider comprises a unique set of categorical parameters. Let $\{C_1, C_2, ..., C_n\}$ denote the set of available cloud providers and $P_i$ the set of all possible node configurations for the $i$-th cloud provider for $i = 1, 2 \ldots, n$. We can then define the functions $f_i(p) : P_i \rightarrow \mathbb{R}$, which, for the $i$-th cloud provider with node configuration $p$, may measure either workload runtime, monetary cost or some other optimization target. We can then formally define the multi-cloud configuration problem as follows:

$$i^* = \arg\min_{i \in [1,n]} \left[ \min_{p \in P_i} f_i(p) \right] \tag{1}$$

We note that this is a joint optimization problem comprising the outer *selection* problem, i.e., which cloud provider to use, as well as the inner *configuration* problem, i.e., how to configure the nodes.

### 2.2 RELATED WORK IN CLOUD LITERATURE

**Configuration subproblem.** There is a significant body of work studying the configuration subproblem: i.e., given a cloud environment $C_i$, how to find the number and types of computing instances that would maximize the performance or minimize the cost of a workload deployment. Chen et al. (2021) present a cloud configuration framework that builds prediction models from small-scale experiments to estimate the performance of experiments on large-scale clusters. Mahgoub et al. (2020) introduce

an online system that adjusts the database and VM configurations as the behavior of the workload changes. Mariani et al. (2017) define a method to estimate the cloud performance of a workload by employing hardware-agnostic application profiles and random-forest-based prediction models. Other prior art from the cloud literature considered employing black-box optimizers (BBOs), such as Bayesian Optimization (BO), to optimize the cloud instance type, cluster size, or both (Yadwadkar et al., 2017a; Alipourfard et al., 2017; Bilal et al., 2020; Hsu et al., 2018b). Additionally, a number of works have looked at improving the performance of BO by augmenting it with low-level performance metrics captured from the workload Hsu et al. (2018a;c).

**Selection subproblem.** None of the approaches mentioned so far have addressed the cloud configuration problem in the *multi-cloud* setting. However, there has been work in the direction of cloud provider selection (without configuration). Chen et al. (2020) introduce a provider selection optimizer (FrugalML) for a specific workload, namely ML inference. Given input data, FrugalML suggests which cloud providers (cloud APIs) should optimally be used for running an inference task so that the user budget is met and the prediction accuracy is increased. Moreover, the selection problem has been widely addressed in the context of multi-cloud service composition (MCSC). Pang et al. (2020) propose a method based on formal concept analysis and skyline hierarchical computation to obtain an optimal MCSC in a mobile edge computing environment. To solve MCSC under uncertain QoS attributes, Haytamy & Omara (2020) describe a method based on Particle Swarm Optimization. Souri et al. (2020) use formal verification to prove the correctness of a MCSC approach that finds the service composition with the minimum number of cloud providers under given QoS user requirements.

**Joint selection-configuration problem.** To the best of our knowledge, the joint cloud selection-configuration problem has been less studied. Yadwadkar et al. (2017b) consider the problem of finding the best VM for a single-node workload across multiple cloud providers. The problem is framed as a *prediction* task and a random forest model is trained on a dataset comprising low-level performance metrics from pre-existing workloads. The trained model can then be used to effectively predict the performance of each VM for unseen workloads. An orthogonal direction, with which our paper is aligned, treats the multi-cloud configuration problem as an *optimization* task and aims to develop algorithms that do not require information collected from pre-existing workloads. Ramamurthy et al. (2020) propose a search-based method that ranks cloud providers for a given workload. The VM configuration for each cloud provider is optimized independently, using an arbitrary optimization algorithm that may take into account memory and/or disk space constraints. Shi et al. (2020) describe a Genetic Algorithm that finds a multi-cloud deployment configuration for multi-service applications. To decrease the complexity of the search, the authors design a clustering algorithm that groups dependent services, so that services in the same cluster are deployed to the same location (provider or region). Challenges arise with this approach when mutations occur that cross cloud providers, since the set of VMs offered by each provider differ significantly.

Figure 1: Relationship between multi-cloud configuration and AutoML.

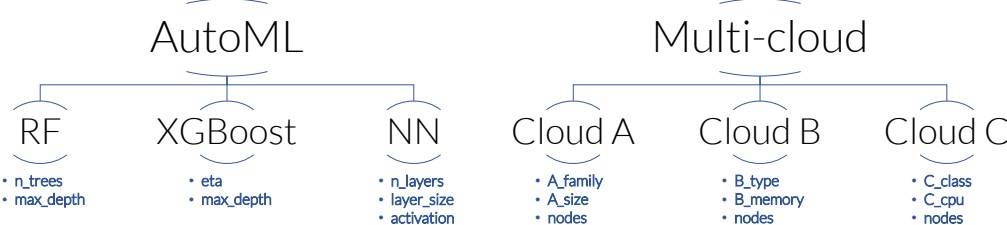

## 2.3 RELATION TO AUTOML

In Bilal et al. (2020) the authors make an analogy between cloud configuration and hyperparameter tuning in machine learning. They point out important differences between these two problems. Firstly, cloud configurations consist mostly of integer or categorical variables, whereas hyperparameters in machine learning are often continuous. Secondly, cloud configurations are typically complex and expensive to evaluate (involving spinning up a cluster of nodes), whereas hyperparameter configurations may be easily evaluated on a single node. Despite these differences, the authors found

that methods that work well for hyperparameter tuning (such as BO with random forests or gradient boosted trees) are also extremely effective for solving the cloud configuration problem.

The key insight of this paper is to go one step further and notice that, due to its hierarchical nature, the selection-configuration problem defined in (1) bears a striking resemblance to the joint optimization problem at the heart of AutoML. Namely, in AutoML one must *select* which ML model to use (e.g., neural networks vs. gradient-boosted decision trees), and decide how to *configure* each model (e.g., how each hyperparameter should be chosen). Figure 1 illustrates this analogy.

### 2.4 RELATED WORK IN AUTOML

Having established this analogy, it makes sense to ask the question: can existing methods from AutoML be applied to the multi-cloud configuration problem (1)? We now proceed to review the state-of-the-art with this question in mind.

Two early, ground-breaking AutoML solutions were auto-sklearn (Feurer et al., 2019) and Auto-WEKA (Thornton et al., 2013). Both of these frameworks remain highly active GitHub projects, and both use sequential model-based algorithm configuration (SMAC) for solving the model selection-configuration problem. SMAC, introduced in Hutter et al. (2011) and recently extended in Lindauer et al. (2021), implements a form of BO that takes into account the natural hierarchy present in selection-configuration problems. SMAC is itself an actively maintained GitHub project, and provides APIs that can be used to solve arbitrary selection-configuration problems (rather than being specific to AutoML). Accordingly, SMAC can be easily applied to multi-cloud configuration.

More recent works have proposed splitting the model selection-configuration problem into simpler subproblems and solving them in an alternating fashion. In principle, all of these approaches can be readily applied to the analogous problem of (1). Firstly, an approach based on the alternating direction method of multipliers (ADMM) was proposed in Liu et al. (2020). Secondly, in Rakotoarison et al. (2019), it was proposed to use Monte-Carlo tree search to solve the selection problem, and BO to solve the configuration problem. Finally, Li et al. (2020) proposed Rising Bandits (RB), an algorithm that treats the model selection problem as a best-arm identification problem, in which each arm corresponds to a ML model and each arm pull corresponds to running BO with a fixed budget to optimize the corresponding model configuration. These methods, in particular RB, served as a main inspiration for the algorithmic approach described in the next section.

On the other hand, there exist AutoML solutions which cannot be applied to multi-cloud configuration in a straightforward manner. For example, frameworks like H2O's AutoML (LeDell & Poirier, 2020) and AutoGluon (Erickson et al., 2020) do not perform *selection*, but rather form ensembles of different, configured ML models. While ensembling has proven very successful in the AutoML context, achieving better accuracy than a single model, it does not have a natural analogy in the context of optimization problem (1). Other AutoML frameworks, such as TPOT (Olson & Moore, 2016), focus more on the problem of feature engineering to achieve high accuracy, rather than selection-configuration, and thus do not relate directly to problem of minimizing runtime and/or cost of cloud workloads. Another popular direction in AutoML are multi-fidelity approaches like Hyperband (Li et al., 2017) or BOHB (Falkner et al., 2018) that try to eliminate bad configurations very quickly by evaluating them using a small resource (for instance, a small subset of the training examples). Such methods cannot be applied to multi-cloud configuration in the general setting since we may not know the specifics about the workload that is to be optimized, and even if we did, it may not admit any sensible notion of resource. Finally, there exist numerous commercial offerings in the AutoML space such as Google Cloud Platform AutoML Tables, IBM AutoAI, H2O Driverless AI, and many more. These proprietary offerings typically provide ML-specific interfaces and the details of how they solve the problem internally remains unknown.

## 3 CLOUDBANDIT - A MULTI-CLOUD CONFIGURATION SOLUTION

### 3.1 ALGORITHM DESCRIPTION

We propose treating the multi-cloud configuration problem as a non-stochastic *best-arm identification problem*, as defined in Jamieson & Talwalkar (2016). Specifically, we have $n$ arms, each arm corresponding to a different cloud provider. Pulling an arm corresponds to running an iteration of

an arbitrary BBO algorithm to find the best configuration for the corresponding cloud provider. Let reward $r_{i,k}$ denote the best runtime or cost obtained by pulling the $i$-th arm $k$ times. Assuming that the reward for the $i$-th arm eventually converges to $\mu_i = \lim_{\tau \to \infty} r_{i,\tau} = \min_{p \in P_i} f_i(p)$, then the multi-cloud configuration problem (1) is equivalent to the problem of identifying $\arg \min_i \mu_i$. To have an efficient algorithm, we would like to achieve this whilst minimizing the total number of arm pulls, henceforth referred to as the *total budget*.

We now describe CloudBandit (CB), a simple algorithm for solving the best-arm identification problem defined above. The algorithm maintains an *active set* of arms $A$, which is initially equal to the complete set of all arms (i.e., all cloud providers). The algorithm then performs a number of rounds equal to the number of providers. The $k$-th round (for $k = 1 \ldots, n$) begins by pulling all arms in the active set $b_k$ times. Next, the arm in the active set with the worst reward is identified and eliminated from the active set. Before proceeding to the next round, the budget is increased by a multiplicative factor: $b_{k+1} = b_k \eta$. At the end of the $n$-th round, the best identified configuration is returned, for the sole remaining provider.

The CB algorithm is defined in full in Algorithm 1. The algorithm has two hyperparameters: the initial budget $b_1$ and the growth factor $\eta$. The total budget of CB can be expressed in terms of these two hyperparameters: $B = \sum_{i=1}^{n} (n-i+1) b_1 \eta^{i-1}$. By using relatively small $b_1$ and setting $\eta > 1$, it is hoped that the algorithm can eliminate slow (or expensive) cloud providers very quickly, whilst exploring the more promising cloud providers exponentially more than those that are eliminated.

---

**Algorithm 1** CloudBandit

1: Initialize set of arms (cloud providers): $A = \{1, 2, \ldots, n\}$ and $\hat{b} = 0$
2: Set initial budget $b_1$ and budget growth factor $\eta$.
3: **for** $k = 1, \ldots, n$ **do**
4:     **for** $i \in A$ **do**
5:         Run $b_k$ iterations of component BBO for provider $i$.
6:         Receive best configuration $p_{i,\hat{b}+b_k}$, and corresponding reward: $r_{i,\hat{b}+b_k} = f_i(p_{i,\hat{b}+b_k})$.
7:     **end for**
8:     Identify the best arm: $i^* = \arg \min_{i \in A} r_{i,\hat{b}+b_k}$.
9:     Eliminate the worst arm: $A = A \setminus \{\arg \max_{i \in A} r_{i,\hat{b}+b_k}\}$
10:    Increment $\hat{b} = \hat{b} + b_k$ and set budget for next round: $b_{k+1} = \eta \cdot b_k$.
11: **end for**
12: Output best provider $i^*$ and its configuration $p_{i^*,\hat{b}}$.

---

We note that, after each round of CloudBandit, it is actually possible to utilize all previously-evaluated configurations by passing them on to the component optimizers in following rounds. However, due to the way most BBO software tools are implemented, this presents quite some technical challenges. We discuss how these challenges can be overcome and the effect of passing this information forward in Appendix B. However, all results in the main manuscript use the vastly simpler approach, where each component optimizer instance is independent.

## 3.2 CLOSELY RELATED ALGORITHMS

**Successive Halving.** A popular method for solving best-arm identification problems is successive halving (SH). It was first proposed in Karnin et al. (2013) for the stochastic setting, in which each arm pull corresponds to sampling from a probability distribution, and later generalized to the non-stochastic setting in Jamieson & Talwalkar (2016). SH is similar to CloudBandit, in that it maintains an active set of arms which are progressively eliminated, and arms that survive are explored exponentially more than arms that don't. The key difference is that SH eliminates a constant fraction $(1/\eta)$ of the arms in each round. While this approach may also be effective in the multi-cloud configuration setting when the number of providers is fairly large, when this number is relatively small (e.g., $n = 3$), it is hard to define a reasonable elimination schedule due to rounding issues.

**Rising Bandits.** In Li et al. (2020), the authors treat the model selection-configuration problem from AutoML as a best-arm identification problem, and propose a new algorithm: Rising Bandits

(RB). In RB, each arm corresponds to a different ML model, and each arm pull corresponds to running a fixed number of iterations of BO to tune the hyperparameters of the corresponding model. The reward is given by the best validation loss found by the optimizer. This algorithm is different to CloudBandit in three important respects. Firstly, each arm pull in RB corresponds to running a number of iterations of BO (rather than an arbitrary BBO). Secondly, in RB, the arms are pulled a fixed number of times in each round (i.e., $b_k$ is constant). Finally, RB does not eliminate arms by simply discarding the arm with the worst reward. Instead, RB makes a theoretical assumption about the way the validation loss converges as a function of BO iterations. Specifically, RB assumes that after a certain number of pulls all arms will reach a point of *diminishing returns*. The authors derive a theoretical criteria for when this point is reached, at which point an arm is eliminated.

## 4    MOCCA: AN OFFLINE MULTI-CLOUD BENCHMARK DATASET

Multi-cloud configuration is an exciting new optimization problem, and we feel that the ML community would be interested in developing new algorithms for this emerging application. However, running online experiments can be time-consuming and costly, making this area of research inaccessible to many in our community. For this reason, offline benchmark datasets that allow researchers to compare different optimization algorithms without performing all the necessary cloud experimentation are highly desirable. While Hsu et al. (2018c) provided an offline dataset regarding configuration of cloud workloads for a single-provider, to the best of our knowledge there are no datasets relating to the multi-cloud scenario. Thus, in order to accelerate our own experimentation, as well as to open this problem up to others, we have constructed MOCCA (Dataset for **M**ulti-**C**loud **C**onfiguration **A**lgorithms) – a multi-cloud benchmark dataset, which will be made publicly available. In this section we will describe some important details regarding how this dataset was constructed.

**Cloud provider and configuration space.** Our experimental data was collected by running workloads on top of Kubernetes clusters, deployed on three different cloud providers. The process of deploying a Kubernetes cluster on each of the providers follows the same general pattern but differs in the details such as provider-specific tooling and authentication. The number of nodes in the cluster is treated as an integer parameter that is common across all providers, and for each provider, we introduce a number of provider-specific categorical variables corresponding the various options (e.g., *family*, *size*, *type*) that are available for configuring the nodes. Throughout the paper, as well as in the dataset itself, the names of the cloud providers (as well as their configuration options) have been anonymized. The total dimensionality of the resulting hierarchical configuration space is 88.

**Workloads.** We consider various ML training and data pre-processing workloads from the `dask-ml` package, a library for scalable ML that runs on top of the Dask framework (Dask Development Team, 2016). For our purposes, Dask workloads were attractive for two main reasons. Firstly, Dask is a distributed framework and workloads can be seamlessly scaled out across nodes. Some workloads may benefit greatly from such scaling, others may suffer in terms of performance as more nodes are added. For this reason, it is typically more difficult to find the optimal configuration of a distributed workload, compared to the single-node case, leading to a more interesting optimization problem for our benchmark. Secondly, Dask has a clean integration with Kubernetes: it is simple to deploy a Dask cluster (using Helm charts) on top of a set of Kubernetes pods. Furthermore, this process is agnostic to where the Kubernetes pods are located, resulting in straightforward scripting even in the multi-cloud setting.

**Optimization tasks.** In our benchmark an *optimization task* comprises: (a) a workload defined as a (Dask task, input dataset) pair, and (b) an optimization target which can be either runtime or cost. We collected data for 30 different workloads (10 Dask tasks running on 3 input datasets) and 2 different optimization targets, leading to a total of 60 different optimization tasks. Details regarding all Dask tasks and datasets can be found in Appendix A. For the runtime target, we recorded the total time taken by a given workload for each cloud provider and configuration, including all data transfer overheads. To generate data for the cost target, we estimate the cost by multiplying the runtime by the listed price-per-hour for each node, as well as a factor equal to the number of nodes. While this is an imperfect estimate, and will not take into account additional data transfer costs, more precise estimates are difficult to obtain since cloud billing typically occurs monthly in an aggregate manner.

# 5 EXPERIMENTAL RESULTS

## 5.1 CHOOSING THE COMPONENT BBO

To begin our experimental evaluation, we will investigate the effect of using different BBOs as components inside CB. The goal of this experiment is twofold. Firstly, we would like to determine which BBO provides the best performance. Secondly, we would like to verify that the performance of a given BBO is consistently improved when used inside CB relative to being naively applied to the *flattened* parameter space. In our experiment we consider four different BBOs: random search (RS), BO with Gaussian processes (GP), BO with random forests (RF) and RBFOpt. While RS is perhaps the simplest baseline, in Li & Talwalkar (2020) it was found to be a surprisingly strong baseline in an AutoML context. BO with a GP surrogate model was proposed in Snoek et al. (2012) for hyperparameter tuning tasks, whereas Bilal et al. (2020) found BO with RF to perform better for the cloud configuration task. In both cases, we use the BO implementation provided by scikit-optimize (Louppe, 2017). RBFOpt is a relatively new BBO and has been shown in Costa & Nannicini (2018) to outperform a wide range of BBOs on a variety of optimization tasks. It is based on the radial basis function method, originally proposed in Gutmann (2001), and was recently improved in Nannicini (2020) to provide better support for categorical variables.

For each BBO, we evaluated its performance both when used as a standalone optimizer, and when used as a component within CB. Experiments were performed for all 30 workloads in the MOCCA dataset, for both time and cost optimization targets. When used inside CB, we used a default growth factor of $\eta = 2$ and varied the value of initial budget $b_1 = 1, \ldots, 8$, which for the $n = 3$ providers in the dataset, results in a total budget of $B = 3b_1 + 2b_1\eta + b_1\eta^2 = 11b_1$ evaluations. In order to ensure a fair comparison in terms of number of evaluations, when used as a standalone optimizer, each BBO was run by varying the budget between $B = 11, 22, \ldots, 88$ evaluations. Default hyperparameters were used for all BBOs, both when used inside CB and when used as a standalone optimizer, with the exception of BO with RF, where the hyperparameters from Bilal et al. (2020) were used. For each workload, optimization target and optimization algorithm, we performed 50 experiments using different random seeds. As an evaluation metric we used the regret: the relative distance to the true minimum, averaged over all seeds.

Figure 2: Probability distribution functions (PDF) for a representative workload.

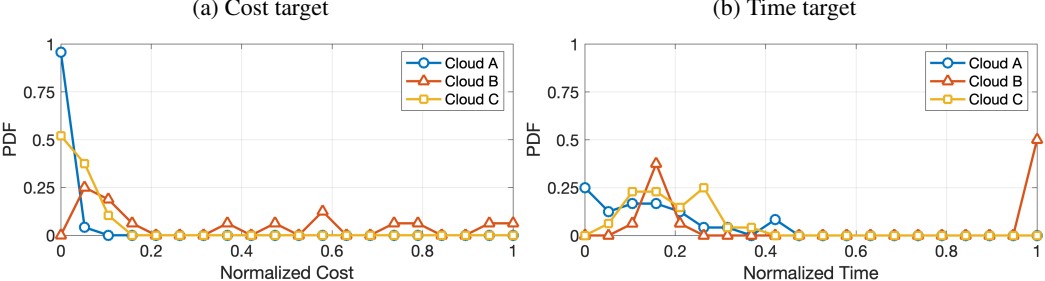

In Table 1 we present the average percentage regret, across all workloads in the MOCCA dataset, as a function of the total number of evaluations. Results are provided for the cost optimization target in Table 1a and for the time optimization target in Table 1b. For each pair of columns corresponding to a given BBO, the lowest regret is highlighted in bold. From these results, we can make a number of observations. Firstly, for all BBOs we observe significantly lower regret when the BBO is used as part of CB, relative to when it is used as a standalone optimizer. On average, the regret is reduced by 20 percentage points (p.p.) for the cost target and by 7 p.p. for time. Secondly, we notice that RBFOpt appears to be the best choice of component BBO for CB, consistently achieving the lowest regret across all budgets. Finally, the best algorithm (CB with RBFOpt) seems to converge to the true minimum faster for the cost target, relative to the time target. We believe this last observation is related to the fact that for the cost target, we typically see one provider that is significantly cheaper that the others, whereas for time the distributions for each provider have more overlap. Statistics are provided in Figure 2, for a representative workload, to illustrate this characteristic. The sensitivity of CB to the overlap between distributions, as well as to the number of cloud providers, has been further investigated using a synthetic dataset in Appendix C.

Table 1: Average percentage regret of CB and component BBOs across all workloads in MOCCA.

(a) Optimization for cost

| budget | RS | | BO with GP | | BO with RF | | RBFOpt | |
|---|---|---|---|---|---|---|---|---|
| | alone | in CB | alone | in CB | alone | in CB | alone | in CB |
| 11 | 87.85 | **48.01** | 120.83 | **54.33** | 93.75 | **53.60** | 66.66 | **48.01** |
| 22 | 47.91 | **30.76** | 38.90 | **33.56** | 70.14 | **27.74** | 32.93 | **18.11** |
| 33 | 33.33 | **18.65** | 25.79 | **18.07** | 62.29 | **18.27** | 17.78 | **6.99** |
| 44 | 24.49 | **13.56** | 20.65 | **12.43** | 56.54 | **12.17** | 12.99 | **0.53** |
| 55 | 18.71 | **7.48** | 18.26 | **9.94** | 51.04 | **9.11** | 9.29 | **0.18** |
| 66 | 14.36 | **4.75** | 16.40 | **8.58** | 46.04 | **6.42** | 5.65 | **0.05** |
| 77 | 11.59 | **2.61** | 15.90 | **7.53** | 43.06 | **4.72** | 4.28 | **0.00** |
| 88 | 8.11 | **2.51** | 15.58 | **7.19** | 40.82 | **3.81** | 2.28 | **0.00** |

(b) Optimization for time

| budget | RS | | BO with GP | | BO with RF | | RBFOpt | |
|---|---|---|---|---|---|---|---|---|
| | alone | in CB | alone | in CB | alone | in CB | alone | in CB |
| 11 | 38.67 | **35.05** | 52.58 | **38.79** | 38.60 | **36.08** | 38.13 | **35.05** |
| 22 | 24.04 | **21.91** | 40.05 | **26.26** | 22.39 | **21.90** | 20.21 | **19.00** |
| 33 | 16.79 | **15.49** | 36.83 | **18.97** | 17.99 | **15.36** | 15.69 | **12.83** |
| 44 | 12.94 | **11.44** | 34.36 | **14.74** | 16.64 | **12.19** | 13.47 | **10.12** |
| 55 | 10.48 | **8.62** | 32.63 | **11.91** | 16.19 | **10.26** | 11.54 | **5.57** |
| 66 | 8.87 | **7.10** | 30.90 | **9.48** | 15.77 | **8.88** | 9.70 | **4.69** |
| 77 | 7.48 | **5.89** | 29.80 | **8.72** | 15.57 | **7.98** | 8.86 | **3.93** |
| 88 | 6.13 | **4.64** | 28.55 | **7.90** | 15.48 | **7.40** | 8.19 | **3.43** |

## 5.2 COMPARISON TO EXISTING AUTOML AND MULTI-CLOUD METHODS

While there have been a number of attempts to solve the multi-cloud configuration problem in the cloud literature, to the best of our knowledge, there is no publicly available code implementing the solutions from either Ramamurthy et al. (2020) or Shi et al. (2020). On the AutoML side, SMAC is a highly active GitHub project, and can be readily applied to our problem and compared with CB. While in principle all of the solutions from Liu et al. (2020); Rakotoarison et al. (2019); Li et al. (2020) could also be used as baselines, none of the papers refer to publicly available implementations. Despite this, we have re-implemented the solutions from Li et al. (2020) (RB) and Ramamurthy et al. (2020) (RM) and included them as baselines.

We thus performed an experiment to compare CB (with RBFOpt as the component BBO) against SMAC, RB and RM. As in the previous experiment, for CB we used a default growth rate of $\eta = 2$ and varied the initial budget $b_1 = 1, 2, \ldots, 8$, leading to a range of total budgets $B = 11, 22, \ldots, 88$. For SMAC and RB we used default hyperparameters. For RM, which allows an arbitrary model to be used to solve the inner configuration task, we use RBFOpt. The budget for all baselines was varied so that the total number of evaluations coincided with the budget that was used for CB. We ran each algorithm with 50 different random seeds, for every workload and target from the MOCCA dataset. For each workload and target, we then computed the average regret across all random seeds.

The percentage regret, averaged over all workloads, is presented in Table 2. Additionally, for each algorithm we present the total accumulated cost (or time) incurred, averaged over all workloads. The cost/time is normalized by the cost/time required for performing an exhaustive search. The winning results are highlighted in bold. From these results we can draw a few interesting conclusions. Firstly, for the cost optimization target, CB out-performs SMAC, RB and RM in terms of regret – on average by 5 p.p., 16 p.p. and 5 p.p. respectively. For larger budgets, SMAC and RM approach the performance of CB in terms of regret, but incur respectively 57% and 56% higher cost on average. In fact, for budget 88, SMAC is equivalent to exhaustive search, since it does not try any configuration more than once. On the other hand, RB is around 45% cheaper than CB on average, but performs significantly worse in terms of regret, suggesting that the theoretical assumptions made by RB may

Table 2: Average regret and normalized accumulated cost/time of CB and baselines.

(a) Optimization for cost

| budget | regret | | | | accumulated cost | | | |
|---|---|---|---|---|---|---|---|---|
| | CB | SMAC | RB | RM | CB | SMAC | RB | RM |
| 11 | **48.0** | 72.1 | 66.7 | 69.0 | **7.8** | 12.5 | 12.0 | 19.1 |
| 22 | **18.1** | 23.0 | 23.1 | 26.4 | 14.7 | 32.5 | **13.8** | 26.5 |
| 33 | **7.0** | 11.8 | 20.1 | 11.9 | 20.4 | 48.7 | **15.3** | 37.9 |
| 44 | **0.5** | 3.1 | 19.3 | 3.1 | 27.1 | 64.7 | **16.7** | 62.8 |
| 55 | **0.2** | 1.8 | 19.1 | 0.5 | 30.6 | 76.6 | **18.0** | 83.8 |
| 66 | **0.0** | 0.2 | 19.1 | 0.1 | 33.7 | 86.8 | **19.4** | 85.6 |
| 77 | **0.0** | 0.1 | 19.1 | **0.0** | 36.2 | 94.5 | **20.8** | 88.1 |
| 88 | **0.0** | **0.0** | 18.8 | **0.0** | 39.1 | 100.0 | **22.1** | 89.2 |

(b) Optimization for time

| budget | regret | | | | accumulated time | | | |
|---|---|---|---|---|---|---|---|---|
| | CB | SMAC | RB | RM | CB | SMAC | RB | RM |
| 11 | **35.0** | 38.1 | 41.9 | 37.9 | **11.9** | 13.0 | 12.9 | 15.0 |
| 22 | 19.0 | 18.7 | 28.4 | **18.5** | 22.8 | 28.5 | **20.2** | 26.6 |
| 33 | **12.8** | 14.7 | 26.1 | 14.3 | 32.7 | 42.7 | **25.5** | 39.4 |
| 44 | 10.1 | **8.0** | 25.1 | 12.2 | 43.0 | 56.2 | **30.5** | 56.4 |
| 55 | 5.6 | **4.5** | 24.0 | 7.1 | 51.9 | 68.5 | **35.3** | 68.2 |
| 66 | 4.7 | **2.3** | 23.5 | 4.4 | 61.0 | 80.1 | **40.0** | 73.7 |
| 77 | 3.9 | **1.1** | 23.0 | 4.3 | 67.3 | 90.3 | **44.6** | 79.7 |
| 88 | 3.4 | **0.0** | 22.5 | 3.5 | 74.2 | 100.0 | **49.2** | 82.5 |

not apply in this new application domain. Secondly, for the time optimization target, the average regret of CB is comparable to that of RM (1 p.p. better on average) but at the same time CB is 18% faster. CB also performs similarly to SMAC in terms of regret for all but the highest budgets (1 p.p. worse on average), whilst being 22% faster. Equivalently to the cost target, RB is 35% faster than CB on average, but in terms of regret, it is 15 p.p. worse on average.

## 5.3 FURTHER EXPERIMENTS

So far, the baseline models in Section 5.2 were run using default hyperparameters. However, the HPs of SMAC and Rising Bandits were set to work well in a different domain. Therefore, we perform a further study of the algorithms' hyperparameters in Appendix D. Furthermore, we consider a scenario in which the user wishes to receive more than one cloud provider recommendation at a time. We analyze and discuss CB's performance in this scenario, as well as a possible modification to the algorithm, in Appendix E. Finally, we consider the dynamic nature of cloud's performance, i.e. dependence on the data center's load. We run experiments on selected workloads repeated multiple times, optimizing various statistical metrics. We present the experimental results in Appendix F.

## 6 CONCLUSION

In this paper, we have presented a key connection between the model selection-configuration problem extensively studied in the AutoML field and the multi-cloud configuration optimization task. Inspired by recent trends in AutoML, we have introduced CB, an algorithm for best-arm identification, that has been specifically designed with the multi-cloud configuration task in mind. To evaluate the performance of CB, we have built and will open to the community a first-of-a-kind benchmark dataset, MOCCA, comprising 60 multi-cloud configuration tasks across 3 public cloud providers. Our experimental results show that CB achieves lower regret on MOCCA relative to a variety of BBOs, and comparable or lower regret relative to state-of-the-art AutoML and multi-cloud solutions, whilst at the same time being either cheaper or faster.

ETHICS STATEMENT

The algorithms described in this paper enable consumers and businesses to deploy large-scale workloads in a multi-cloud setting in a more cost-efficient way. In the particular case of ML-based workloads, we acknowledge that societal concerns exist regarding safety, privacy, fairness and more. These concerns are often valid, and we acknowledge that helping to scale ML workloads further and further may have an indirect, potentially negative, impact on the way ML models are used in society. On the positive side, we feel that helping people optimize their computing workloads across multiple cloud providers can have a positive economic impact: helping to prevent the emergence of a large, all-powerful monopoly in the ever-growing cloud industry.

REPRODUCIBILITY STATEMENT

The MOCCA dataset and code to reproduce the tables from the paper have been prepared and are ready to be shared publicly. We feel strongly about sharing these materials, so that others in the community can reproduce and improve upon our results. Unfortunately, our institution does not permit us to share code publicly (e.g. on OpenReview or GitHub) without a clear indication of authorship. If the paper is accepted, we will upload both the dataset and code, to a repository on public GitHub.

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

## A    DETAILS OF THE MOCCA DATASET

In Table 3 we present the details of Dask tasks, input datasets and optimization targets included in the MOCCA dataset.

Table 3: Details of the optimization tasks used in the dataset.

| Dask tasks | | Datasets | Optimization targets |
|---|---|---|---|
| • kmeans | • polynomial features | • santander (Kaggle, 2019) | • cost |
| • linear regression | • quantile transformer | • credit card (Kaggle, 2018) | • runtime |
| • logistic regression | • spectral clustering | • buzz in social media | |
| • naive bayes | • standard scaler | (Kawala et al., 2013) | |
| • poisson regression | • xgboost | | |

## B    CONTINUOUS CLOUDBANDIT

In Section 3 we have described the CloudBandit algorithm which may come in two variants: continuous (in each round, CloudBandit utilizes all previously-evaluated configurations by passing them on to the component BBO) or intermittent (the component optimizer is independent in each round and does not utilize any of the previously-evaluated configurations). In Section 5 we have presented the experimental results for the latter variant. In this section we will present the results for the continuous variant too.

### B.1    IMPLEMENTATION CHALLENGES

It is necessary to note that depending on the BBO used, sometimes implementing the continuous variant of CloudBandit may be difficult or impossible. Some BBOs, e.g. Random Search, are conceptually unable to utilize any information about past configuration evaluations at all. In other

cases, using continuous CB can be impossible due to the optimizer's incompatible API, which e.g. does not allow the user to pass previous evaluations or to extract all evaluations from the current round.

In cases when the BBO's API is incompatible, it may be possible to find a workaround. An example of such a case is RBFOpt, which does not allow the user to extract all the configurations evaluated by the optimizer. They can only be extracted in an encoded form used internally by the optimizer. In order to adjust RBFOpt to be used with the continuous CloudBandit scheme, one can use an additional wrapper that was developed for the NeurIPS 2020 BBO Challenge [1]. The wrapper utilizes synchronization primitives in order to pause and resume the optimization process, feeding one evaluation at a time. This way, all configurations can be collected and passed to the optimizer in later CB rounds.

## B.2    COMPARISON AGAINST INTERMITTENT CLOUDBANDIT

Table 4: Average ranks of continuous and intermittent CloudBandit.

(a) Optimization for cost

| budget | CB BO with GP | | CB BO with RF | | CB RBFOpt | |
|---|---|---|---|---|---|---|
| | intermittent | continuous | intermittent | continuous | intermittent | continuous |
| 11 | 3.23 | **3.13** | **4.13** | 4.67 | 3.33 | **1.60** |
| 22 | **4.13** | **4.13** | **3.70** | 4.83 | 2.20 | **1.00** |
| 33 | 4.00 | **3.03** | **4.97** | 5.70 | 1.83 | **1.00** |
| 44 | 3.50 | **2.67** | **4.90** | 5.80 | 1.27 | **1.00** |
| 55 | 3.33 | **2.80** | **4.87** | 5.67 | 1.13 | **1.00** |
| 66 | 3.37 | **2.50** | **4.63** | 5.73 | **1.07** | 1.10 |
| 77 | 3.03 | **2.53** | **4.83** | 5.53 | **1.10** | **1.10** |
| 88 | 3.03 | **2.47** | **4.90** | 5.23 | **1.10** | 1.13 |

(b) Optimization for time

| budget | CB BO with GP | | CB BO with RF | | CB RBFOpt | |
|---|---|---|---|---|---|---|
| | intermittent | continuous | intermittent | continuous | intermittent | continuous |
| 11 | **4.00** | **4.00** | 3.30 | **3.07** | **2.73** | 3.03 |
| 22 | 4.40 | **4.33** | **3.77** | **3.77** | 2.10 | **1.77** |
| 33 | 4.87 | **4.50** | **3.73** | 4.17 | 2.23 | **1.50** |
| 44 | 4.67 | **3.83** | **3.97** | 5.00 | 1.87 | **1.53** |
| 55 | 4.30 | **3.73** | **4.17** | 5.13 | **1.70** | 1.73 |
| 66 | **3.73** | 3.83 | **4.20** | 5.53 | 1.77 | **1.57** |
| 77 | 3.73 | **3.37** | **4.30** | 5.57 | **1.50** | 1.53 |
| 88 | 3.37 | **3.10** | **4.43** | 5.57 | 1.47 | **1.40** |

In Tables 4a and 4b we present the average ranks of the continuous CloudBandit scheme compared against the corresponding intermittent scheme for different component BBOs. The ranks were determined based on all algorithms' results averaged over 50 random seeds and all workloads in the MOCCA dataset. Random Search was excluded as a component BBO in this study, since intermittent and continuous CloudBandit are identical by construction. Best results within each pair of continuous and intermittent CloudBandit with the same BBO are highlighted in bold.

When comparing a number of optimization algorithms across a large collection of tasks, rather than applying parametric statistical tests (such as Student's t-test) on a per-task basis, Demšar (2006) argues that it is preferable to perform non-parametric tests across the collection.

In this case, we have a family of 6 optimization algorithms ranked across 60 different multi-cloud configuration tasks (30 workloads with cost optimization target and 30 workloads with time optimization target), for 8 different budgets. In order to verify that statistical differences exist within the family,

---

[1] https://bbochallenge.com/

Table 5: Pairwise p-values comparing continuous vs. intermittent CB.

(a) Optimization for cost

| Budget | Intermittent vs. Continuous CB | | |
| --- | --- | --- | --- |
| | BO with GP | BO with RF | RBFOpt |
| 11 | 2.20e-01 | 6.02e-02 | 1.09e-03 |
| 22 | 5.00e-01 | 1.38e-03 | 1.73e-06 |
| 33 | 5.18e-04 | 1.02e-03 | 1.24e-05 |
| 44 | 9.90e-05 | 4.87e-05 | 4.49e-02 |
| 55 | 2.95e-02 | 1.94e-03 | 1.85e-01 |
| 66 | 2.31e-03 | 2.59e-04 | 4.96e-01 |
| 77 | 1.87e-02 | 8.65e-03 | 5.00e-01 |
| 88 | 2.35e-02 | 2.26e-01 | 4.96e-01 |

(b) Optimization for time

| Budget | Intermittent vs. Continuous CB | | |
| --- | --- | --- | --- |
| | BO with GP | BO with RF | RBFOpt |
| 11 | 5.00e-01 | 2.62e-01 | 4.26e-01 |
| 22 | 3.11e-01 | 3.83e-01 | 1.66e-01 |
| 33 | 8.53e-02 | 6.93e-02 | 1.63e-02 |
| 44 | 2.40e-03 | 1.55e-03 | 2.23e-01 |
| 55 | 1.98e-02 | 1.85e-03 | 4.24e-01 |
| 66 | 3.37e-01 | 1.21e-04 | 1.86e-01 |
| 77 | 1.18e-01 | 1.57e-05 | 4.79e-01 |
| 88 | 2.24e-01 | 1.85e-05 | 3.87e-01 |

we first apply the correction of the Friedman omnibus test proposed in Iman & Davenport (1980). We were able to firmly reject the family-wise null hypothesis (via the omnibus test) with $p < 0.004$ for all budgets, and both time and cost optimization targets. Secondly, we perform pairwise testing using the Wilcoxon signed-rank test (Benavoli et al., 2016) (correcting for multiple hypotheses via the procedure defined in Li (2008)) to verify if differences exist between the optimization algorithms themselves. The pairwise p-values are presented in Table 5a and Table 5b. The cases for which the null hypothesis can't be rejected ($p < 0.05$) are highlighted in red.

When using BO with GP as the component BBO, we find that the continuous variant improves the average rank in a majority of cases. For the cost-based target, the p-values suggest that this improvement is statistically significant, whereas for the time-based target the null hypothesis cannot be rejected most of the time. For BO with RF, we find that the continuous variant results in worse performance relative to the intermittent variant. Furthermore, the p-values suggest this worsening is statistically significant. One explanation for this could be that for BO with RF the number of initial random evaluations in the component BBO was set to 3 following the best hyperparameters identified in Bilal et al. (2020), whereas for BO with GP the number of initial random evaluations is set to 10. It may be that using fewer random evaluations results in less *exploration* and more *exploitation*, leading to convergence to local minima, and the continuous variant amplifies this effect. Finally, for RBFOpt with a time-based target, while we see the average rank improve in many cases, the p-values suggest these differences are not statistically significant. However, for the cost-based target we do see a statistically significant improvement in the average rank for lower budgets.

In conclusion, while we do see an improvement from the continuous variant in some scenarios, there are others where it clearly makes things worse, and others still where statistical differences do not seem to be present. Since significant technical work may be needed to get the continuous variant working for some component BBOs (as described in Section B.1), we opted to present the simpler intermittent scheme in the main manuscript.

## C    SENSITIVITY ANALYSIS

We performed an analysis of how sensitive CloudBandit's results are, both to the search space distribution, as well as the number of providers. In those experiments, we compared ranks of standalone Random Search and CloudBandit using Random Search as the component BBO, calculated by averaging results over multiple random seeds. To be able to simulate various scenarios, the sensitivity experiments were conducted using a synthetic dataset. Each provider was assumed to produce random results drawn from a normal distribution $\mathcal{N}(\mu_i, \sigma^2)$.

Table 6: Average ranks of RS and CB with RS for various scenarios run on a synthetic dataset.

(a) Varying standard deviation $\sigma$

| $\sigma$ | Random Search | |
| --- | --- | --- |
| | alone | in CB |
| 1000 | 1.34 | 1.33 |
| 100 | 1.34 | 1.33 |
| 10 | 1.38 | 1.33 |
| 5 | 1.41 | 1.31 |
| 2 | 1.49 | 1.24 |
| 1 | 1.54 | 1.18 |
| 0.1 | 1.56 | 1.16 |

(b) Varying number of providers $n$

| $n$ | Random Search | |
| --- | --- | --- |
| | alone | in CB |
| 12 | 1.84 | 1.08 |
| 10 | 1.81 | 1.10 |
| 8 | 1.78 | 1.11 |
| 6 | 1.71 | 1.14 |
| 5 | 1.68 | 1.15 |
| 4 | 1.62 | 1.17 |
| 3 | 1.54 | 1.18 |
| 2 | 1.39 | 1.18 |

### C.1    SEARCH SPACE DISTRIBUTION

In this experiment, we assume $n = 3$ providers and the means of the providers' distributions were fixed as $\mu_i = i$ for $i = 0, \ldots, n - 1$. We were then able to manipulate the overlap between providers' distributions by changing the standard deviation $\sigma$ of the distributions. In Table 6a we present the average ranks of CB with RS and standalone RS for various standard deviations $\sigma$. As expected, for small values of $\sigma$, i.e., when the distributions have almost no overlap, CloudBandit gives clearly better results than standalone Random Search. For big values of $\sigma$, where the distributions significantly overlap, we can observe that CloudBandit still wins over standalone RS. However, the difference between average ranks of the two methods is less significant than for small values of $\sigma$.

The results presented in Table 6a allow us to explain why CloudBandit gives better results on the MOCCA dataset when optimizing for cost than in case of optimizing for time, as presented in Tables 1a and 1b. As presented in Figures 2a and 2b, in practice the providers' distributions show bigger overlap for runtime measurements than in terms of cost.

### C.2    NUMBER OF PROVIDERS

We further evaluated CloudBandit's results for various numbers of providers $n$. In this experiment, each case has a radically different search space size because of the varying number of providers. Therefore, instead of using the same total budget for each test case, we fix the initial budget of CloudBandit to $b_1 = 8$. The average ranks of CloudBandit with RS and standalone RS are presented in Table 6b. In cases where many providers were available, CloudBandit has a significant advantage over standalone Random Search. The difference between the methods decreases with the decline in the number of providers, as each arm rejection in CloudBandit carries ever higher risk. However, even for only 2 providers, CloudBandit gives better results than Random Search.

It should be noted that CloudBandit's complexity grows rapidly with the growing number of providers. A possible solution to this issue could be to reject more than one provider at the end of each round in order to reduce the number of rounds needed. This possible modification could motivate future work on CloudBandit.

## D    HYPERPARAMETER TUNING

In the previous experiment, default hyperparameters (HPs) were used for both AutoML baselines: SMAC and RB, and a default growth rate of $\eta = 2$ was used for CB, alongside default HPs for the component RBFOpt. Since the defaults for SMAC and RB were determined for a different application, it is necessary to check whether their performance, as well as that of CB, can be improved by HP tuning.

To achieve this, we made a 67%/33% validation/test split of the workloads from MOCCA. For each of the three algorithms, we defined a HP grid of equal size. The budget for all algorithms was fixed to 33. We then performed a grid search to optimize the HPs: identifying the HP configuration that achieved the lowest average regret on the validation set. For each algorithm, the best configuration was then evaluated on the test set.

Table 7: Average regret using default and tuned hyperparameters.

| optimization target | HP setting | CB | | SMAC | | RB | |
|---|---|---|---|---|---|---|---|
| | | val | test | val | test | val | test |
| cost | best | 1.20 | 0.19 | 5.14 | 25.05 | 10.02 | 30.95 |
| | default | 3.50 | 13.96 | 5.14 | 25.05 | 10.77 | 38.74 |
| time | best | 11.69 | 11.52 | 12.64 | 11.75 | 20.23 | 22.13 |
| | default | 13.48 | 11.53 | 14.60 | 14.84 | 26.04 | 26.23 |

Table 8: Detailed test results for CloudBandit and AutoML algorithms.

| workload | Optimization for cost | | | Optimization for time | | |
|---|---|---|---|---|---|---|
| | CB | SMAC | RB | CB | SMAC | RB |
| buzz-polynomial_features | 1.88 | **0.47** | 1.49 | **0.35** | 1.48 | 6.11 |
| buzz-spectral_clustering | **0.00** | 0.49 | 1.92 | 7.91 | **2.54** | 17.70 |
| buzz-xgboost | **0.00** | **0.00** | 0.39 | 88.97 | **86.86** | 138.81 |
| creditcard-naive_bayes | **0.00** | 4.28 | 22.04 | 6.18 | **5.41** | 9.77 |
| creditcard-standard_scaler | **0.00** | 24.79 | 20.80 | **0.92** | 2.89 | 4.71 |
| santander-kmeans | **0.00** | 3.23 | 3.19 | **1.57** | 3.90 | 7.46 |
| santander-linear_regression | **0.00** | 1.44 | 8.41 | **1.28** | 2.17 | 4.77 |
| santander-poisson_regression | **0.00** | 195.28 | 239.53 | **1.71** | 1.94 | 4.34 |
| santander-quantile_transformer | **0.00** | 20.53 | 11.31 | **0.68** | 2.44 | 3.57 |
| santander-xgboost | **0.00** | **0.00** | 0.39 | **5.59** | 7.90 | 24.01 |

Table 9: Hyperparameter search space for each of the compared algorithms.

| algorithm | parameter | values | best setting for cost | best setting for time |
|---|---|---|---|---|
| CB | $b_1$ | 1, 2, 3, 4, 5 | 1 | 3 |
| | rbf | cubic, linear, gaussian | linear | gaussian |
| | algorithm | Gutmann, MSRSM | Gutmann | Gutmann |
| SMAC | mode | SMAC4HPO, SMAC4AC | SMAC4HPO | SMAC4AC |
| | acq_opt_challengers | 50, 500, 5000 | 500 | 500 |
| | rand_prob | 0.1, 0.3, 0.5, 0.7, 0.9 | 0.5 | 0.5 |
| RB | estimator | RF, GP | GP | RF |
| | n_init | 2, 5, 10, 12, 15 | 15 | 15 |
| | acq_func | LCB, PI, EI | PI | EI |

The results are presented in Table 7 which shows the average regret, on both the validation set and test set, for each of the three algorithms, using both the default and tuned HPs. We see that, in most cases,

HP tuning helped to improve the average regret on the test set, relative to the default HPs. However, there is no change in the relative order of the algorithms, both for the cost, as well as for the time optimization target. Additionally, in Table 8 we list the average test scores of best HP configurations of all 3 algorithms, individually for each test workload. The winning results for each workload and optimization target were highlighted in bold. Finally, Table 9 lists all hyperparameters tuned for each algorithm, together with value ranges and best configurations selected for both optimization targets.

## E    BEST-$k$ ARMS SELECTION

Another extension of CloudBandit is a scenario in which the user wishes to receive more than one cloud recommendation at a time and choose themself which provider they wish to use. Specifically, we focus on a case in which $k = 2$ best providers (and their best configurations) are to be identified. This can be achieved by running the traditional 1-best-arm identification CloudBandit scheme as presented in Section 3.1 and suggesting 2 best providers found throughout the optimization process (the arms that survived up to round $n - 1$). However, this approach treats the suggested arms unfairly, as one of the arms receives a significant additional evaluation budget in the final round. To solve this issue, a simple and effective modification of CloudBandit for this scenario can be the 2-best-arms scheme, which stops the algorithm when the size of the set of active arms $A$ is down to $k = 2$ arms, so that both candidates are treated equally budget-wise.

In order to evaluate CloudBandit's performance in this scenario, we run both schemes described above on the MOCCA dataset, this time expecting 2 arm candidates each time. We ensure fair comparison by setting equal total evaluation budgets (identical starting budget $b_1$ and different growth factors $\eta = 2$, $\eta = 4$ for 1-best-arm and 2-best-arms schemes, respectively). Table 10 presents the average regrets of both the traditional 1-best-arm CB scheme and the 2-best-arms scheme. The regrets of the first (better) candidates (denoted as option 1 in Table 10) were calculated with respect to the true minimum of the MOCCA dataset, similarly to Section 5. The regrets of the second (worse) candidates (denoted as option 2) were calculated with respect to the minimum over all providers except for the one which gives the true minimum. The better result across CB schemes for each candidate is highlighted in bold. The results for both optimization targets were averaged over 50 random seeds and all 30 workloads. As the component BBO, we use RBFOpt in both cases.

Table 10: Average regrets of 2 best provider candidates for 1-best-arm and 2-best-arms CB schemes.

| budget | Optimization for cost | | | | Optimization for time | | | |
|---|---|---|---|---|---|---|---|---|
| | 2-best-arms | | 1-best-arm | | 2-best-arms | | 1-best-arm | |
| | option 1 | option 2 | option 1 | option 2 | option 1 | option 2 | option 1 | option 2 |
| 11 | 58.74 | **96.33** | **48.01** | 128.77 | **34.79** | **46.56** | 35.05 | 63.29 |
| 22 | 21.70 | **21.21** | **18.11** | 72.61 | **18.59** | **24.17** | 19.00 | 38.27 |
| 33 | 9.03 | **15.06** | **6.99** | 32.94 | 13.59 | **16.91** | **12.83** | 26.33 |
| 44 | 0.64 | **11.50** | **0.53** | 17.05 | **9.82** | **13.84** | 10.12 | 19.41 |
| 55 | **0.18** | **6.20** | **0.18** | 11.33 | **5.34** | **11.10** | 5.57 | 15.00 |
| 66 | **0.00** | **2.39** | 0.05 | 9.22 | **4.18** | **8.20** | 4.69 | 11.40 |
| 77 | **0.00** | **0.88** | **0.00** | 8.18 | **3.41** | **6.48** | 3.93 | 8.89 |
| 88 | **0.00** | **0.41** | **0.00** | 7.21 | **2.51** | **4.73** | 3.43 | 7.07 |

The 2-best-arms scheme gives much better second candidates for both optimization targets, which was expected, as this scheme devotes a bigger budget to the second best arm. For the cost optimization target, the 1-best-arm scheme produces comparable or better first candidates than the 2-best-arms scheme for most budgets. However, for the runtime optimization target, the 2-best-arms scheme produces better first candidates than the 1-best-arm scheme, although it gives a smaller evaluation budget to the winning arm. As it was shown in Figure 2b, there is a significant overlap between providers' distributions for the runtime optimization target. Better results of the 2-best-arms scheme for this optimization target may indicate that when there is too much overlap between providers' distributions, the regular 1-best-arm CloudBandit scheme can sometimes reject providers too aggressively. In such cases, CloudBandit may therefore benefit from keeping more than one provider in the final optimization round.

# F DYNAMIC EFFECTS

One of important issues in the cloud is the dynamic nature of performance, which may differ depending on the load on the data center at any given instant. The tail effects (worst case scenario results) are of special interest from the users' perspective, so that their workloads are guaranteed not to exceed some acceptable limits.

To analyze CloudBandit's performance in the dynamic environment, we re-run some of the workloads from MOCCA multiple times. For this experiment, we chose the 5 fastest workloads, as they should be the most susceptible to tail effects. We consider various statistical metrics that can be used by CloudBandit as the optimization metric: the mean, median and 90th percentile of results (as used by Yadwadkar et al. (2017b)). The experiments were run for the time optimization target and repeated over multiple random seeds.

Table 11 compares the average regrets of CloudBandit with RBFOpt and standalone RBFOpt for the analyzed statistical metrics. Additionally, Figures 3a-3c present the probability distribution functions (PDFs) of the analyzed metrics of all 3 cloud providers for a representative workload. We can see that despite differences between the PDFs of the analyzed statistical metrics, CloudBandit's results are comparable for all of them. This confirms that CloudBandit can be used universally, regardless of the metric of interest.

Figure 3: PDFs of various statistical metrics for a representative workload.

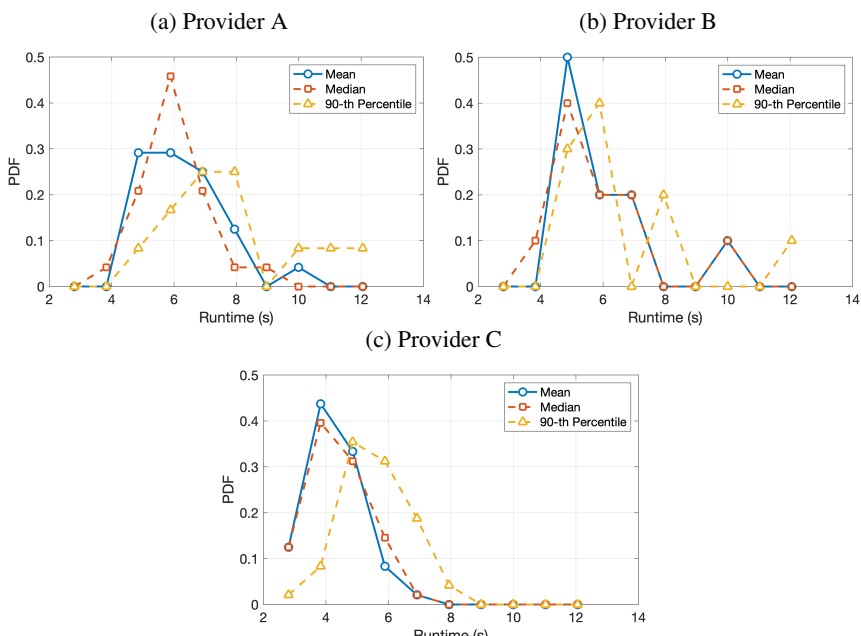

Table 11: Average regrets of CloudBandit using various statistical metrics of the data.

| budget | mean | | median | | 90th percentile | |
|---|---|---|---|---|---|---|
| | RBFOpt | CB | RBFOpt | CB | RBFOpt | CB |
| 11 | 21.45 | **17.25** | 19.98 | **16.54** | 23.53 | **19.59** |
| 22 | 12.72 | **8.03** | 11.25 | **8.51** | 14.18 | **8.81** |
| 33 | 8.79 | **6.24** | 8.42 | **5.97** | 10.64 | **6.74** |
| 44 | 7.04 | **4.20** | 6.97 | **4.66** | 8.57 | **4.83** |
| 55 | 6.06 | **3.81** | 5.39 | **3.80** | 7.25 | **4.21** |
| 66 | 5.29 | **2.69** | 4.45 | **3.01** | 5.99 | **3.23** |
| 77 | 4.23 | **2.11** | 3.88 | **2.01** | 5.17 | **1.98** |
| 88 | 3.86 | **1.67** | 3.33 | **1.42** | 4.41 | **1.45** |

