# OpenReview forum: "What can multi-cloud configuration learn from AutoML?"
_ICLR.cc/2022/Conference — ICLR 2022 Submitted_

### Official Review · Reviewer_FdKV · 2021-10-17

**Correctness:** 2
**Technical Novelty And Significance:** 2
**Empirical Novelty And Significance:** 3
**Recommendation:** 5
**Confidence:** 4

**Main Review:**

The pros: 1) The authors firstly build a benchmark for the multi-cloud configuration task. 2) they evaluate the proposed methods comprehensively, compared to BBOs and other AutoML methods.
The cons and questions：1) Can you justify the dataset is correct? Because it is an imperfect estimate, and the more precise estimates are difficult to obtain for the cost target. Does the dataset consider the different prices/costs from different providers? It seems to matter and impact the correctness of the proposed dataset for the cost target.
2）Does the CB algorithm output only one best arm/provider? In practice, the users might choose multi-cloud to avoid vendor lock-in. Does it seem the goal of the algorithm should output top-n best arms/providers?
3）The situation for AutoML and Multi-cloud configuration seems to be different in Fig. 1. In the Multi-cloud configuration case, each provider provides the same type of parameters, e.g., CPU number/type，memory size，VM/container type. But for AutoML，the parameter types are totally different for different models.  Is the special character in the multi-cloud configuration case helpful to solve the multi-cloud configuration problem?

**Summary Of The Paper:**

In this paper, the authors analyze the similarity between the multi-cloud configuration optimization task and the model selection-configuration problem, and try to solve the multi-cloud configuration optimization task inspired by an algorithm for best-arm identification from the AutoML domain.  They built a benchmark for the multi-cloud configuration task including  60 multi-cloud configuration tasks across 3 public cloud providers. The experimental results show the proposed method can find best cloud provider with best instances and other parameters more cheaper or faster, compared to BBOs.

**Summary Of The Review:**

In this paper, the authors try to solve the multi-cloud configuration problem with an algorithm from the AutoML domain. It looks good, but it requires more explanation and clarification why it is a good/best way by the proposed method, why it is better than other methods, why it works. And it also requires more justification of the correctness of the evaluation data.

---

> ### Author Response · Authors · 2021-11-19
> **Author Response to Reviewer FdKV**
>
> Thank you for the feedback and valuable suggestions. Below we have addressed the questions and concerns from the review.
>
> **Correctness of Dataset**
>
> Our dataset comprises for two statistics for every workload across every different (provider-configuration)-pair: one for runtime and one for estimated cost. The runtime statistics is 100% accurate because we have really deployed the workload and measured it - there is no estimation involved here. Obtaining a perfect measurement of the cost of each evaluation is a difficult problem, since billing occurs in aggregate at the end of each month. Thus it is necessary to form some estimate. We follow the same approach as [PARIS] and [Scout], and estimate the cost as the total runtime multiplied by the number of nodes multiplied by the published cost for each node. This method is defined in the paper in Section 4 - "Optimization tasks". We accept that this in an imprecise estimate, and does not take into account effects like data transfer costs. Obtaining more accurate estimates is an interesting direction, however we feel strongly that this should not be a barrier to publication. In particular, the [Scout dataset](https://github.com/oxhead/scout), which uses the same approach for single-cloud configuration, has been used in many research papers [Arrow, Scout, Micky, Bilal].
>
> **Correlations between cloud configuration parameters**
>
> Each cloud provider can be configured using a unique set of categorical variables. While these categories are different for all providers, it is indeed true that they are somehow correlated since under-the-hood they define the CPU manufacturer/generation, memory size and number of cores. We think it is a very interesting idea to try to exploit these correlations and this could be a promising research direction. We note that to a lesser-extent such correlations may also be present in the AutoML setting (e.g. the hyperparameters of XGBoost and LightGBM and CatBoost are very closely related).
>
> **Top-k arms**
>
> Our work was developed under the assumption that CloudBandit would be run periodically (e.g. once a week/month) to reevaluate which cloud provider/configuration should be used currently for a given workload. For this reason, in all our experiments so far, CloudBandit's goal was to suggest a single best (cloud provider, configuration)-pair. However, we absolutely agree that the case in which the user wishes to receive more than one candidate cloud provider (+ its configuration) is important to consider and we thank the reviewer for suggesting this idea. To evaluate CloudBandit in this scenario, we considered 2 possible approaches: one, in which CloudBandit runs as usual and returns k=2 providers instead of 1, and another, in which CloudBandit has the same total evaluation budget but stops rejecting arms when the set of active arms is down to k=2. Our experiments show that this k-best-arms modification of CloudBandit works better than the 1-best-arm scheme and that CloudBandit can be successfully used in the k-best-arms scenario as well. We present and discuss the results of both schemes in Appendix E.
>
> [PARIS] "Selecting the Best VM across Multiple Public Clouds: A Data-Driven Performance Modeling Approach." N. J. Yadwadkar, B. Hariharan, J. E. Gonzalez, B. Smith and R. H. Katz, SoCC'17.
>
> [Scout] "Scout: An Experienced Guide to Find the Best Cloud Configuration".  C-J. Hsu, V. Nair, T. Menzies and V-W. Freeh, arXiv, 2018.
>
> [Arrow]  "Arrow: Low-Level Augmented Bayesian Optimization for Finding the Best Cloud VM". C. Hsu, V. Nair, V. W. Freeh and T. Menzies, ICDCS'18.
>
> [Micky] "Micky: A Cheaper Alternative for Selecting Cloud Instances." C. Hsu, V. Nair, T. Menzies and V. Freeh, CLOUD'18.
>
> [Bilal] "Do the best cloud configurations grow on trees? An experimental evaluation of black box algorithms for optimizing cloud workloads." M. Bilal, M. Serafini, M. Canini, and R. Rodrigues, VLDB'20.

---

> > ### Author Response · Authors · 2021-12-01
> > **Follow up**
> >
> > Thank you again for your feedback on our paper. We would like to kindly ask if you have any additional comments on the updated manuscript, in particular regarding the additional experiments (finding the top-k cloud providers) that we performed based on your recommendation.

---

### Official Review · Reviewer_5pqw · 2021-10-24

**Correctness:** 4
**Technical Novelty And Significance:** 2
**Empirical Novelty And Significance:** 2
**Recommendation:** 5
**Confidence:** 4

**Main Review:**

The main contributions of the paper are: 1) A method to choose the best cloud provider in a multi cloud scenario to avoid vendor lock in given a task. The method provides an objective metric and helps narrow down the choice based on runtime or cost. 2) A dataset for benchmarking and comparisons (although this has to be released publicly yet). and 3) A comprehensive study against other Black box optimizers is provided. The optimization is studied firstly to choose which BBO is better for the search internally in CB and then it is evaluated against SMAC, RB to compare the overall algo.  Main concerns are the following 1) CB is evaluated over runtime and cost currently. However, in practice there are many variables in play such as region availability, SLA requirements, number of instance availability, networking requirements etc., Without actually looking at the current 60 multi cloud configurations that it has been evaluated on, it is hard to comment on the how the tasks were setup and how generally applicable this is or add more configurations. 2) Given that cloud providers provide custom rates for long term usage under corporate deals, in practice the cost estimation can vary quite a lot and passing these custom configurations to CloudBandit seems to be unknown.

**Summary Of The Paper:**

The paper proposes a method to find the best cloud provider for multi cloud compute selection (time, cost) to avoid vendor lock in. They achieve this using a best-arm identification algorithm where each arm runs an BBO on each provider and eventually choosing the best arm. They test this approach on 60 different configuration across 3 providers using a proposed dataset and show that the algorithm provides cheaper, faster config. given a set of parameters.

**Summary Of The Review:**

The authors study the problem of multi-cloud configuration in this paper, where the problem can be defined as how to choose the best cloud provider based on runtime or cost for a given workload. They propose an algorithm (CB) to help solve this and compare the algorithm for its effectiveness against SMAC, RB and show that it performs well at solving the optimization problem. More work needs to be done to show how complex workloads can be translated into configs that CB can solve and also account for more parameters when optimizing for runtime or cost as discussed above.

---

> ### Author Response · Authors · 2021-11-18
> **Author Response to Reviewer 5pqw**
>
> We thank the reviewer for their feedback. In the following paragraphs we would like to address the concerns outlined in the review.
>
> **General Applicability of CloudBandit and adding more configurations**
>
> CloudBandit can be applied to any workload that runs on top of Kubernetes. The user provides as an input the workload, and receives as an output a (provider, configuration)-pair. CloudBandit is an online (or search-based) algorithm: it searches through the hierarchical provider-configuration domain, iteratively evaluating (provider-configuration)-pairs and receiving a reward (runtime or estimated cost), to find the pair that provides the best reward. The user is free to define which cloud providers they would like to use, as well as the node configuration domain for each provider. Additionally, if the user has negotiated a special pricing offer with a particular cloud provider, this information can easily be incorporated by adjusting how the cost is estimated when performing evaluations.
>
> **Details about the offline dataset**
>
> In order to design and simulate the performance of CloudBandit without racking up enormous cloud bills, we have developed an offline dataset (MOCCA). In order to construct the dataset we first defined a provider-configuration domain consisting of 88 different cloud configurations spanning 3 different cloud providers. We have not provided details regarding these configurations since it would effectively de-anonymize the MOCCA dataset, which we would like to publish upon acceptance. The anonymity of the dataset is important to us, since we would like the dataset to be used purely for academic research rather than for making competitive analysis between cloud providers. Next, we defined 30 different workloads, which were picked to be both relevant to the ML community and heterogeneous in terms of the kinds of computation and communication required (e.g. tree-building with XGBoost vs.  ADMM for logistic regression). More details are provided in Table 3 in the Appendix. We then proceeded to run each workload on every (provider, configuration)-pair in the domain, collecting both runtime and estimated cost statistics.  Effectively, this created a table with 60 different columns (30 columns correspond to the runtime measurement for each workload, and the other 30 for the cost estimated), and 88 rows (each row corresponding to a different (provider, configuration)-pair).
>
>
> This offline dataset is used to simulate the behaviour of online optimization algorithms. In the (fully) online scenario, at each round, the algorithm would select a set of cloud configurations from the user-defined config space (which can be anything the cloud providers support), run the workload on those selected configurations and measure the performance/ estimate the cost. Instead, using the offline dataset, at each round, the algorithm selects the configurations and only reads the performance/cost already measured/estimated for those configurations. The goal of the algorithm is essentially to find the row containing the lowest runtime/cost for every column. This is equivalent to finding the configuration with the lowest runtime/cost in a (fully) online scenario.
>
> **Taking into account additional constraints**
>
> We chose to focus on optimizing for runtime and cost because there is a large body of work [CherryPick, Arrow, Micky, Scout] that treats the single-cloud configuration problem in the same way. We feel that our paper should be viewed as an extension of these works in the multi-cloud direction. Adding more complex effects (such as, region availability, SLA requirements) is an exciting future research direction.
>
>
> [CherryPick] "CherryPick: Adaptively Unearthing the Best Cloud Configurations for Big Data Analytics." O. Alipourfard, H. Harry Liu, J. Chen, S. Venkataraman, M. Yu, and M. Zhang. NSDI'17
>
> [Arrow]  "Arrow: Low-Level Augmented Bayesian Optimization for Finding the Best Cloud VM". C. Hsu, V. Nair, V. W. Freeh and T. Menzies, ICDCS'18
>
> [Micky] "Micky: A Cheaper Alternative for Selecting Cloud Instances." C. Hsu, V. Nair, T. Menzies and V. Freeh, CLOUD'18.
>
> [Scout] "Scout: An Experienced Guide to Find the Best Cloud Configuration".  C-J. Hsu, V. Nair, T. Menzies and V-W. Freeh, arXiv, 2018

---

> > ### Author Response · Authors · 2021-12-01
> > **Follow up**
> >
> > Thank you again for your feedback on our paper. We would like to kindly ask whether our response above resolved any of your concerns.

---

### Official Review · Reviewer_mcUn · 2021-11-02

**Correctness:** 4
**Technical Novelty And Significance:** 2
**Empirical Novelty And Significance:** 3
**Recommendation:** 5
**Confidence:** 5

**Main Review:**

Overall, the motivation and idea presented in the paper is clear. However, it is significantly missing in details wrt to datasets, experiments and cloud provider configuration space.

I would implore authors to think about following limitations of the work that should be addressed (and possibly their future work):

1. Taskset: Tasks in the current work only consist of ML-workloads which are algorithmically fixed, i.e., there is little to no control flow/branching.

2. Dynamic environment: How will the results change across the cloud providers if the cloud itself is heavily loaded. Tail latency is a big issue in the cloud environment. See "Selecting the Best VM across Multiple Public Clouds: A Data-Driven Performance Modeling Approach" published in SOCC 17.

3. Moving away from black-box assumption: There is no requirement in this task to be workload agnostic, i.e., the users of the CloudBandit should have access to the workload and should be able to leverage insights from workload details (such as memory required, code complexity, processor, etc). One can obtain such details by profiling and tracing of the code and execution environment. Roofline models (see https://en.wikipedia.org/wiki/Roofline_model) should also help model aspects of the code and execution environment to significant reduce the accumulated cost while providing great performance.

**Summary Of The Paper:**

This pager presents CloudBandit to solve the multi-cloud selection-configuration problem. While the problem is heavily explored in several domains, this paper leverages the hierarchical aspect of the selection-configuration problem. The main novelty of the paper lies in creating the dataset for the above problem. Such dataset can be immensely useful for academic research as "executing code" in cloud can be prohibitively expensive in academic settings; thereby, hindering the academic research.

**Summary Of The Review:**

Overall, the paper is well written. However, several aspects of the problem were ignored (see main review). The dataset presented could be meaningful but currently it lacks the rigor in generating the dataset or providing meaningful motivation for selecting "dask" tasks. This paper has the seed for significant contribution but the current quality of work is not sufficient for publication in a conference.

---

> ### Author Response · Authors · 2021-11-19
> **Author Response to Reviewer mcUn**
>
> We would like to thank the reviewer for their feedback. Below we address the questions and concerns raised in the review.
>
> **Details missing regarding datasets, experiments and config space**
>
> In Table 3 in the Appendix we have provided details regarding the workloads that were run, as well as the input datasets that were used for each workload. Details regarding the config space have not been revealed since this would de-anonymize the identity of the cloud providers. This anonymity is important to us, since we would like to release the MOCCA dataset publicly upon publication and we would like to avoid legal issues that may arise if the dataset can be used to make claims that one cloud provider is "better" than another.
>
> **ML workloads are algorithmically fixed and have little to no control flow / branching**
>
> The workloads we have used are not algorithmically fixed. The details of the workloads we have used were provided in Table 3 in the Appendix and include a variety of different ML training and data pre-processing workloads. Training of an XGBoost model involves building a large number of trees, resulting in plenty of branching logic and bearing little-to-no resemblance to the kinds of operations needed to, for example, train a Logistic Regression model or perform Spectral Clustering.
>
> **Dynamic environment**
>
> We agree that dynamic effects in the cloud are important and thank the reviewer for pointing us to this paper. We have included the suggested reference [PARIS] in our related work section.  To capture the cloud tail effects, the solution described in this reference work uses the 90th percentile of the run-times. The optimization metric used by CloudBandit can be any statistical metric, incl. the 90th percentile. The results included in the initial version of our paper were for workloads run only once, mainly due to research budget limitations. However, in the meantime, we managed to re-run some of the workloads (the top-5 fastest workloads that should be the most susceptible to tail effects) multiple times to capture such effects. We then re-ran CloudBandit for the mean, median and 90th percentile of the collected run-times. We present and discuss the results in Appendix F. From the conducted experiments we can conclude that CloudBandit's performance does not depend significantly on the statistical metric used to capture the cloud's dynamic nature and can be used universally, regardless of the user's choice of metric.
>
> **Moving away from the black box assumption**
>
> It is correct that a number of works [Arrow, Scout] have shown that it is possible to improve the performance of Bayesian Optimization for the single-cloud configuration problem by including low-level metrics. CloudBandit could indeed be generalized to support component optimizers that leverage such metrics, and this is an interesting research direction and next step. On the other hand, there exist plenty of published works that do not use such information [CherryPick, Micky, Bilal, Shi] and treat single or multi-cloud configuration as a pure black-box optimization problem. The primary goal of our paper is to validate the hypothesis that such black-box optimization methods can benefit from acknowledging the additional hierarchy that exists in the multi-cloud setting. We feel that our results make a very convincing case for this, and that this is an important contribution to the field. The fact that CB might be improved by taking into account low-level metrics does not invalidate our contributions.
>
>
> [PARIS] "Selecting the Best VM across Multiple Public Clouds: A Data-Driven Performance Modeling Approach." N. J. Yadwadkar, B. Hariharan, J. E. Gonzalez, B. Smith and R. H. Katz, SoCC'17.
>
> [CherryPick] "CherryPick: Adaptively Unearthing the Best Cloud Configurations for Big Data Analytics." O. Alipourfard, H. Harry Liu, J. Chen, S. Venkataraman, M. Yu, and M. Zhang. NSDI'17
>
> [Micky] "Micky: A Cheaper Alternative for Selecting Cloud Instances." C. Hsu, V. Nair, T. Menzies and V. Freeh, CLOUD'18.
>
> [Arrow]  "Arrow: Low-Level Augmented Bayesian Optimization for Finding the Best Cloud VM". C. Hsu, V. Nair, V. W. Freeh and T. Menzies, ICDCS'18.
>
> [Scout] "Scout: An Experienced Guide to Find the Best Cloud Configuration".  C-J. Hsu, V. Nair, T. Menzies and V-W. Freeh, arXiv, 2018.
>
> [Bilal] "Do the best cloud configurations grow on trees? An experimental evaluation of black box algorithms for optimizing cloud workloads." M. Bilal, M. Serafini, M. Canini, and R. Rodrigues, VLDB'20.
>
> [Shi] "Location-aware and budget-constrained service deployment for composite applications in multi-cloud environment." T. Shi, H. Ma, G. Chen, and S. Hartmann.  IEEE Transactions on Parallel and Distributed Systems'20.

---

> > ### Comment · Reviewer_mcUn · 2021-11-29
> > **Thank your for the discussion**
> >
> > Thank you for addressing the comments in detail and running those additional experiments. However, it is still not clear to me if the dataset and the algorithms are significant.
> >
> > First, if the tail latency is not a problem then the common sense tells me that not all providers are equally competitive. I am having hard time believing that fact. Authors may be seeing these effects because: (1) selection bias in choosing cloud providers, (2) selection bias in choosing configuration. As a reviewer, it is hard to assess that without access to the configuration parameters.
> >
> > In similar vain, if the dynamics are so stationary (as mentioned by authors in some other comment --- "algorithms are supposed to run once per mont") and there is a clear winner (i.e., a provider which always does the best), then would not an experts be able to choose the right configuration using their prior knowledge. If this is not the case, the paper should reflect the non-intuitive answers given by the algorithm compared to an expert. Result should also highlight, why one provider or config was chosen compared to another. Cloud provider names are already anonymized so giving these kind of details should be okay.
> >
> > I will update my score accordingly.

---

### Official Review · Reviewer_bhwc · 2021-11-04

**Correctness:** 3
**Technical Novelty And Significance:** 2
**Empirical Novelty And Significance:** 2
**Recommendation:** 5
**Confidence:** 3

**Main Review:**

Strengths
=========
1. The paper discusses a direction of research which will become more and more relevant in the years to come. As such this paper may inspire future researchers to work on the problem of multi-cloud configuration which will enable customers to minimize runtime and cost while running workloads on clouds provided by multiple vendors, without explicitly having to choose one vendor over another.
2. The paper draws an analogy to AutoML and uses some of the concepts to come up with a novel approach based on best-arm identification problem. This is one of the key contributions of the paper.
3.  Another major contribution of the paper is the release of the multi-cloud configuration dataset which will be made publicly available. In line with my comment above this will further enable researchers to work on this area and make further inroads.
4. The experimental section appears very detailed. The authors have compared the use of different BBOs (black box optimizers) inside CB. The comparison has been done for both cost and runtime targets. However the 2nd part of the experimental section has some weakness as described under “weaknesses”.

Weaknesses
===========
1. The main algorithm for optimisation viz. CloudBandit appears to be a simple extension of existing approaches and does not seem to have enough novelty (considering the exclusiveness of ICLR)
2. Although there are multiple similar approaches the authors have run experiments against only 2 baselines viz. SMAC and RB. The authors argue that other approaches cannot be compared against as there is no public implementations available.Thus it not clear how this method compares against the SOTA approaches.



**Summary Of The Paper:**

he paper  provides a solution for multi-cloud configuration problem. Specifically, the paper tries to provide cloud customers with an optimal configuration to minimise runtime and cost. The paper also presents a dataset, for offline benchmarking, comprising of 60 different multi-cloud configuration tasks across 3 cloud service providers.

**Summary Of The Review:**

The authors have done a commendable job in a very relevant research area which will become more prominent in the coming years. The key contributions are: 1) CloudBandit a best-arm identification problem for multi-cloud configuration optimization. and 2) the public release of a multi-cloud configuration dataset.
However the novelty does not seem to be adequate keeping in mind the standards of the conference. Also there are some weaknesses in the evaluation results and the comparision with SOTA is not clear.

---

> ### Author Response · Authors · 2021-11-19
> **Author Response to Reviewer bhwc**
>
> We thank the reviewer for the feedback. In the following paragraphs we address the concerns raised in the review.
>
> **Novelty of CB**
>
> We have taken care to put CloudBandit properly in the context of related algorithms from the literature in Section 3.2.  We feel that the differences to both Successive Halving and Rising Bandits have been clearly outlined, and with the latter we have included a comprehensive performance comparison. While we accept as fair criticism that these differences may not warrant a paper in themselves, our paper has multiple other contributions. In particular, we are the first to propose this connection between AutoML and multi-cloud configuration, and we will provide the MOCCA dataset to enable other researchers to build and improve upon our work in this area.
>
> **New baseline**
>
> Since submission of the presented manuscript, we were able to implement one more baseline, this time from the multi-cloud configuration field: the work by Ramamurthy et al. [Ramamurthy]. The work acknowledges the hierarchy of the search space but does not reject any of the providers early on. Instead, all providers are treated equally in terms of budget. The work suggests that any arbitrary model can be used to solve the inner configuration problem, so in our implementation we used RBFOpt, which performed best in case of CloudBandit. We added the new baseline to Section 5.2 of the manuscript. From the results of our experiments we conclude that CloudBandit performs better in terms of regret, relative to the model proposed in [Ramamurthy] for the cost optimization target and comparably for time optimization target. At the same time, CloudBandit is 56% cheaper when optimizing for cost and 22% faster when optimizing for time.
>
> [Ramamurthy] "Selection of cloud service providers for hosting web applications in a multi-cloud environment." A. Ramamurthy, S. Saurabh, M. Gharote and S. Lodha, IEEE SCC'20.

---

> ### Author Response · Authors · 2021-12-02
> **Follow up**
>
> Thank you again for your feedback on our paper. We would like to kindly ask if you have any additional comments on the updated manuscript, in particular regarding the additional baseline that we added based on your recommendation.

---

### Decision · Program_Chairs · 2022-01-20

**Decision:**

Reject

**Comment:**

This paper studies the problem of choosing the best cloud provider for a task. The problem is formulated as a bandit and solved using algorithm CloudBandit. The algorithm is compared to several baselines, such as SMAC, and performs well. The evaluation is done on 60 different multi-cloud configuration tasks across 3 public cloud providers, which the authors want to share with the public.

This paper has four borderline reject reviews. All reviewers agree that it studies an important problem and that the promised multi-cloud optimization dataset could spark more research in the area of cloud optimization. The weaknesses of the paper are that it is not technically strong and that the quality of the new dataset is not clear from its description. At the end, the scores of this paper are not good enough for acceptance. Therefore, it is rejected.